# Hexagonal WO₃·0.33H₂O Hierarchical Microstructure with Efficient Photocatalytic Degradation Activity

**Wei Li** [1,2,†], **Tingting Wang** [1,†], **Dongdong Huang** [1], **Chan Zheng** [1,2,*], **Yuekun Lai** [3,*], **Xueqing Xiao** [1], **Shuguang Cai** [1] and **Wenzhe Chen** [1]

1   School of Materials Science and Engineering, Fujian University of Technology, 3 Xueyuan Road, Fuzhou 350108, China; liwei@fjut.edu.cn (W.L.); WangTingTing9502@163.com (T.W.); hdd20201027@163.com (D.H.); 19781148@fjut.edu.cn (X.X.); sgcai@fjut.edu.cn (S.C.); chenwz@fjut.edu.cn (W.C.)
2   Institute of Materials Surface Technology, Fujian University of Technology, 3 Xueyuan Road, Fuzhou 350108, China
3   College of Chemical Engineering, Fuzhou University, Fuzhou 350116, China
*   Correspondence: czheng@fjut.edu.cn or czheng.fjut@gmail.com (C.Z.); yklai@fzu.edu.cn (Y.L.); Tel.: +86-591-22863283 (C.Z. & Y.L.)
†   These authors contributed equally.

**Abstract:** Structural design and morphological control of semiconductors is considered to be one of the most effective ways to improve their photocatalytic degradation properties. In the present work, a hexagonal WO₃·0.33H₂O hierarchical microstructure (HWHMS) composed of nanorods was successfully prepared by the hydrothermal method. The morphology of the HWHMS was confirmed by field-emission scanning electron microscopy, and X-ray diffraction, Raman spectroscopy, and thermogravimetric analysis demonstrated that the synthesized product was orthorhombic WO₃·0.33H₂O. Owing to the unique hierarchical microstructure, the HWHMS showed larger Brunauer–Emmett–Teller (BET) surface and narrower bandgap (1.53 eV) than the isolated WO₃·0.33H₂O nanorods. Furthermore, the HWHMS showed enhanced photocatalytic activity for degradation of methylene blue under visible-light irradiation compared with the isolated nanorods, which can be ascribed to the narrower bandgap, larger BET specific surface area, and orthorhombic phase structure of the HWHMS. This work provides a potential protocol for construction of tungsten trioxide counterparts and materials similar to tungsten trioxide for application in gas sensors, photocatalysts, electrochromic devices, field-emission devices, and solar-energy devices.

**Keywords:** tungsten trioxide hydrate; hierarchical structure; band gap; specific surface area; photocatalytic property

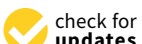

## 1. Introduction

With the continuous growth of industrialization, increasing energy consumption and the corresponding environmental pollution problems have attracted increasing and widespread attention. In particular, comprehensive treatment of industrial wastewater has become a hot topic in the environmental science community. Several conventional methods have been used to treat wastewater, including biological treatment, anaerobic, aerobic, and electrochemical methods, oxidation, reduction, flotation, flocculation, precipitation, and adsorption. However, these methods cannot completely degrade organic pollutants [1–6]. Photocatalysis based on illumination of semiconductor powder for removal of organic contaminants from water is environmentally friendly and cost effective, and it has; therefore, attracted particular interest. Titanium dioxide (TiO₂) is the most extensively studied photocatalyst for decontamination of water. Unfortunately, its wide band gap (about 3.30 eV) contributes to its low conversion rate in the visible light region, greatly limiting its application in the field of photocatalysis [7–10].

Tungsten trioxide ($WO_3$) is a transition-metal-oxide semiconductor with a band-gap energy ranging between 2.4 and 3.0 eV at room temperature. Hence, $WO_3$ has strong visible-light adsorption, contrary to other photocatalysts, such as $TiO_2$, which have light absorption in the harmful ultraviolet (UV) spectral region owing to their inherent band gap energies [11–13]. In addition to the favorable region of light absorption, $WO_3$ is resistant to photocorrosion and has stable physicochemical properties [14,15]. These properties make $WO_3$ a promising alternative to $TiO_2$, which needs to be modified to absorb visible light. In addition to its favorable photocatalytic properties, the $WO_3$ surfaces possess highly negative surface charges that are ideal for adsorption applications, especially for adsorption of cationic dyes, such as methylene blue (MB) [16,17]. Previous studies have revealed that the degradation rate of organic materials is mainly related to the amount of catalytic surface active sites, which depends on the surface area, light utilization, and structural properties of the catalyst, including the phase composition, crystallinity, size distribution, and morphology [18–20]. Among these properties, morphology control has attracted attention because the photocatalytic properties of $WO_3$ can be effectively tuned by tailoring the shape and dimensionality in practical applications.

The hierarchical architectures of semiconductors have attracted great attention because of their unique properties, owing to the well-organized morphology, porous structures, and high surface-to-volume ratio and permeability compared with conventional nanocrystallites. Consequently, much effort has been devoted to synthesis of $WO_3$ hierarchical structures and investigation of their photocatalytic activities for degradation of organic dyes [21–25], particularly for low-dimensional building blocks, such as nanoparticles, nanorods, and nanoflakes. For example, hierarchical hollow $WO_3$ shells containing dendrites, spheres, and dumbbells have been synthesized [21]. Compared with commercial $WO_3$ particles, all the obtained hollow shells with large Brunauer–Emmett–Teller (BET) surface areas showed enhanced photocatalytic activities for degradation of organic contaminants under visible-light irradiation. Nanoflake-based hollow sphere-like and flower-like hydrated tungsten oxide three-dimensional (3D) architectures have been successfully synthesized by a facile chemical solution route without any surfactants [22]. Both of the structures exhibited good abilities for degradation of rhodamine B (RhB) under 365-nm UV light and visible light. A facile solvothermal method has also been developed to synthesize 3D microdahlia $WO_3·0.33H_2O$ hierarchical structures by adding acetone without any additives [23]. After 240-min photodegradation, 88.5% of RhB was eliminated by the hierarchical microdahlia $WO_3·0.33H_2O$. However, for practical application in photocatalysis, fabrication of desired hierarchical micro/nanostructures is very important because of their unique structures and properties. It is still challenging to organize $WO_3$ nanoscale building blocks into well-defined two-dimensional (2D) and 3D micro/nanostructures with high performance to meet the requirements of practical applications.

The photocatalytic properties of tungsten trioxide hydrate ($WO_3·nH_2O$, $n$ = 0–2) with a hierarchical structure have also been reported, and a few studies have reported preparation of self-assembled $WO_3·0.33H_2O$ hierarchical microstructures. Here, we fabricated of a novel well-defined hexagonal $WO_3·0.33H_2O$ hierarchical microstructure (HWHMS) with high uniformity and investigated its photocatalytic performance for degradation of organic contaminants. The morphology, composition, structure, and linear optical characteristics of the HWHMS were investigated in detail by field-emission scanning electron microscopy (FESEM), X-ray diffraction (XRD), Raman spectroscopy, thermogravimetric analysis (TGA), and UV–Visible spectroscopy. The possible growth mechanism of the $WO_3·0.33H_2O$ hierarchical structures is also proposed. The HWHMS showed enhanced photocatalytic activity for degradation of MB under visible-light irradiation compared with isolated $WO_3·0.33H_2O$ nanorods, which can be ascribed to the narrower bandgap, larger BET specific surface area, and orthorhombic phase structure of the HWHMS.

## 2. Results and Discussion

### 2.1. Characterization of the HWHMS

The HWHMS and the precursor material as well as the literature were synthesized by the hydrothermal method. The phase purity and crystallographic structure of the sample were investigated by XRD (Figure 1). By comparing PDF standard card, the main substance of precursor was determined to be $WO_3 \cdot H_2O$ (PDF# 43-0679). All of the diffraction peaks of the as-synthesized HWHMS can be exclusively indexed to orthorhombic $WO_3 \cdot 0.33H_2O$ (JCPDS Card No. 54-1012), with lattice constants of $a = 12.5271$ Å, $b = 7.7367$ Å, and $c = 7.3447$ Å. Furthermore, all of the main diffraction peaks of the HWHMS were relatively strong and narrow, confirming the high crystallinity.

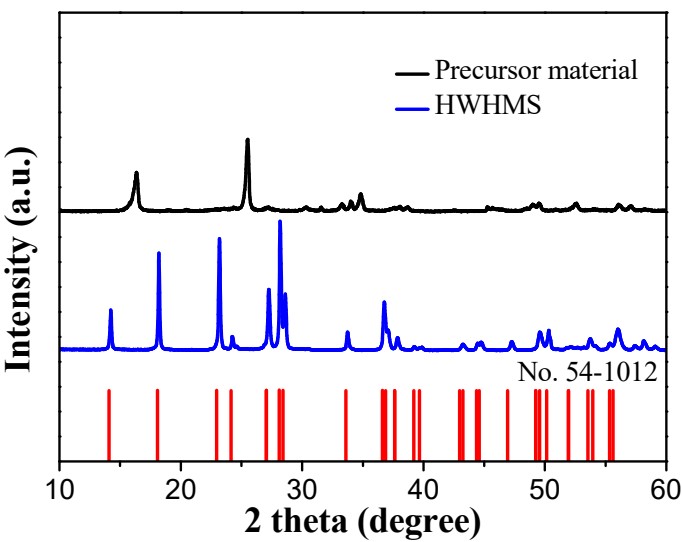

**Figure 1.** XRD pattern of as-synthesized HWHMS and the precursor material.

To investigate the morphology of the as-prepared sample, we performed FESEM measurements. A typical SEM image of the precursor material and obtained HWHMS are shown in Figure 2a,b, respectively. From the clear outline in Figure, the HWHMS exhibited a hierarchically hexagonal morphology with uniform shape and size. The side length of a single HWHMS was approximately 1.2 μm and the thickness was about 80 nm. A high-resolution SEM image (Figure 2b) showed that the HWHMS was mainly composed and stacked by nanorods, which were ordered and some were face-to-face aligned. The growth mechanism is proposed in Figure 2c. First, hydrochloric acid and sodium tungstate interact and form pale-yellow $H_2WO_4$ seeds. Subsequently, when hydrogen peroxide ($H_2O_2$) solution is added, the precipitate dissolves. The reactants are then subjected to surface tension during the recrystallization process, which is attributed to the excess $H_2O_2$. The hexagonal structure has a stable surface energy and; therefore, the nanorods with high surface energy agglomerate in the direction where the Gibbs free energy decreases. Similarly, Li et al. [24] reported synthesis of $Ni(OH)_2$ hexagonal platelets that were self-grown on three-dimensional Ni foam by one-step hydrothermal treatment of Ni foam in 15 wt% $H_2O_2$ aqueous solution.

The composition and structure of the synthesized HWHMS were further characterized by Raman spectroscopy and TGA. Raman spectroscopy can determine the molecular vibration and rotation information by capturing the scattering information, which is sensitive to the structure. The Raman spectrum of the HWHMS is shown in Figure 3a. Six vibration peaks located at 169.5, 196.5, 277.5, 333.4, 692.1, and 802.0 $cm^{-1}$ can be observed in the Raman spectrum. According to related literature [25,26], the peaks at 169.5 and 196.5 $cm^{-1}$ are because of internal lattice vibration of the product. The vibration peaks at around 277.5 and 333.4 $cm^{-1}$ correspond to bending vibration of W–O–W. The vibration peak at 333.4 $cm^{-1}$ is attributed to tensile vibration of W–OH$_2$. In addition, the strong Raman vibration peaks at 692.1 and 802.0 $cm^{-1}$ can be attributed to tensile vibration of O–W–O.

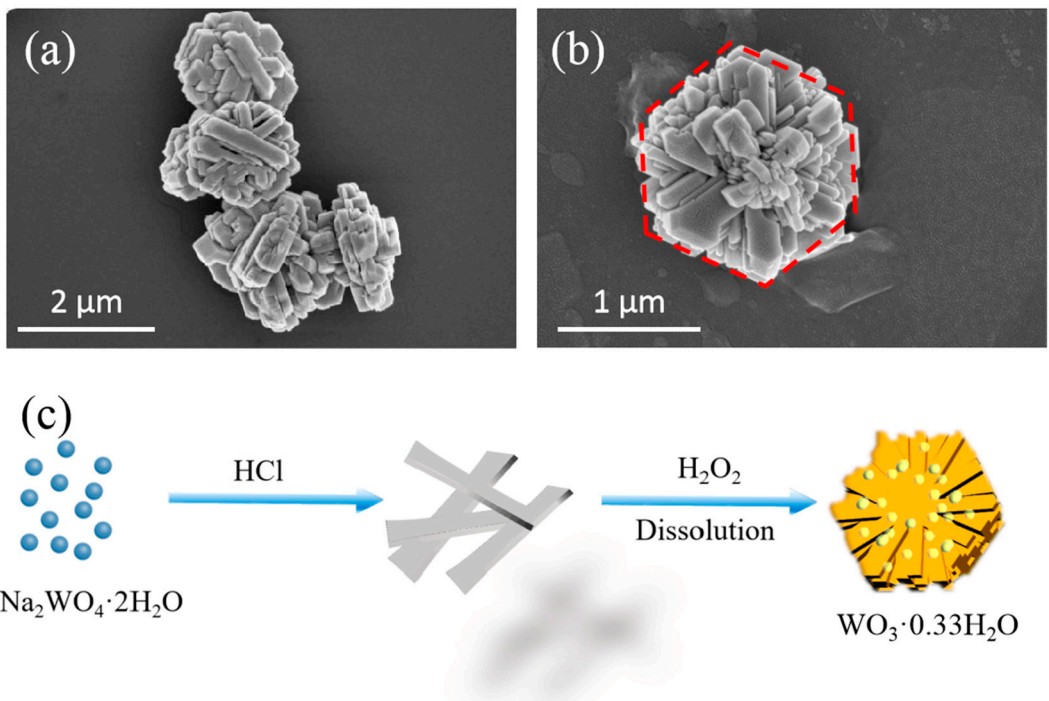

**Figure 2.** Typical SEM images of (**a**,**b**) HWHMS and (**c**) proposed growth mechanism of the HWHMS.

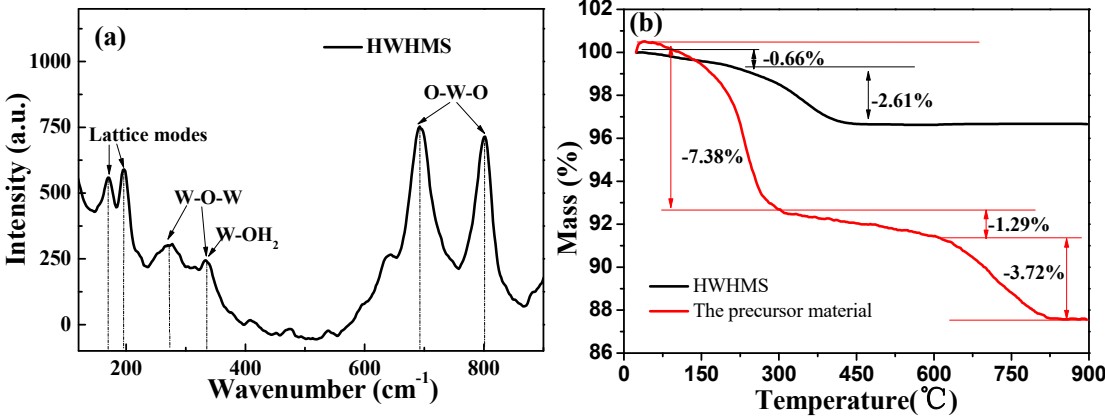

**Figure 3.** (**a**) Raman spectrum of HWHMS; (**b**) TG curve of HWHMS and the precursor material.

The thermal stability and chemical components were investigated by TGA (Figure 3b). The weight loss process of the as-prepared HWHMS can be divided into two stages. The first slight weight loss stage (about 0.60%) of the sample occurred between 0 to 150 °C, which is attributed to evaporation of free water from the HWHMS. The second weight loss stage occurred from 150 to 430 °C, and it was more obvious than the previous stage. The weight loss in the second stage was probably associated with the loss of combined water. The weight loss of 2.61% in the second stage was relatively close to the calculated weight loss of 2.52% based on the loss of $H_2O$ from the $WO_3 \cdot 0.33H_2O$ precipitate. The weight loss process of the precursor material can be divided into three stages. The first slight weight loss stage (about 7.38%) of the sample occurred between 0 to 300 °C, which is attributed to evaporation of free water from precursor material. The second slight weight loss stage (about 1.29%) of the sample occurred between 300 to 600 °C, which is attributed to thermal decomposition of solvents from the precursor material. The third weight loss stage occurred from 600 to 800 °C the weightlessness of the third stage is 3.72%, and it was probably associated with the loss of combined water. The weight loss rate

of the precursors was more pronounced than that of HWHMS. All of the above results confirmed formation of the well-defined HWHMS through assembly of nanorods by a facile hydrothermal method.

The texture and pore structure of the resulting HWHMS were investigated by nitrogen adsorption–desorption. The results are shown in Figure 4a and the Barrett–Joyner–Halenda pore size distribution is shown in Figure 4b. The BET surface area of the HWHMS was 17.6181 $m^2/g$, which is favorable for application in photocatalytic degradation of organic pollutants. The HWHMS possessed a broad and bimodal pore-size distribution with small (40–50 Å) and large (200–500 Å) mesopores. The small mesopores reflect the porosity within the nanorods, while the larger mesopore are related to the pores formed between the stacked nanorods. The HWHMS with many different sized mesopores could reduce transport limitation in catalysis, and it is suitable for harvesting the photoenergy and transport of reactive molecules [27,28].

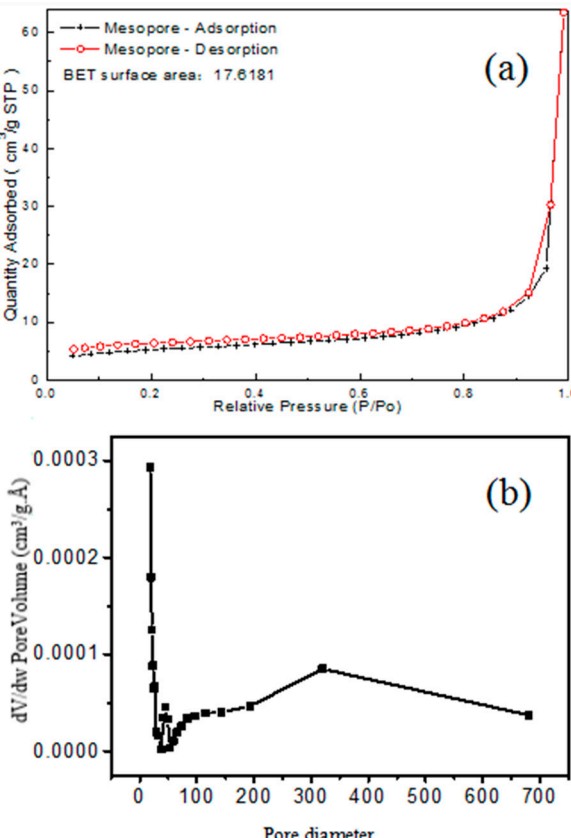

**Figure 4.** (**a**) Typical $N_2$-gas adsorption–desorption isotherms and (**b**) the corresponding pore-size distribution of the HWHMS.

## 2.2. Optical Properties of the HWHMS

The optical properties of the HWHMS were investigated by UV–Vis absorption spectroscopy, and the results are shown in Figure 5a. The weak signal at about 554 nm, which is in the visible region, can be assigned to the absorption peak of $WO_3$. The absorption intensity of the sample decreased as the wavelength of incident light blue shifted. The Kubelka–Munk conversion equation was used to calculate the band-gap energy of the HWHMS [29]:

$$\alpha h\upsilon = A(h\upsilon - E_g)^{n/2}, \tag{1}$$

where $\alpha$, $E_g$, and $A$ are the absorption coefficient, band-gap energy, and general constant, respectively. $h\upsilon$ is the photon energy. $n$ depends on the transition properties of the semiconductor ($n = 1$ for an allowed direct transition and $n = 4$ for an allowed indirect transition). For $WO_3 \cdot 0.33H_2O$, the transition is indirect. To further explore its band

structure, the band-gap energies can be estimated from a plot of $(\alpha h\nu)^{1/2}$ versus photo-energy ($h\nu$) and the intercept of the tangent to the plot will give an approximation of the indirect band-gap energies of the samples. The plots of $(\alpha h\nu)^{1/2}$ versus $h\nu$ are presented in Figure 5. The band gap energies of the precursor material and the HWHMS are calculated to be 2. 41 and 3.32 eV.

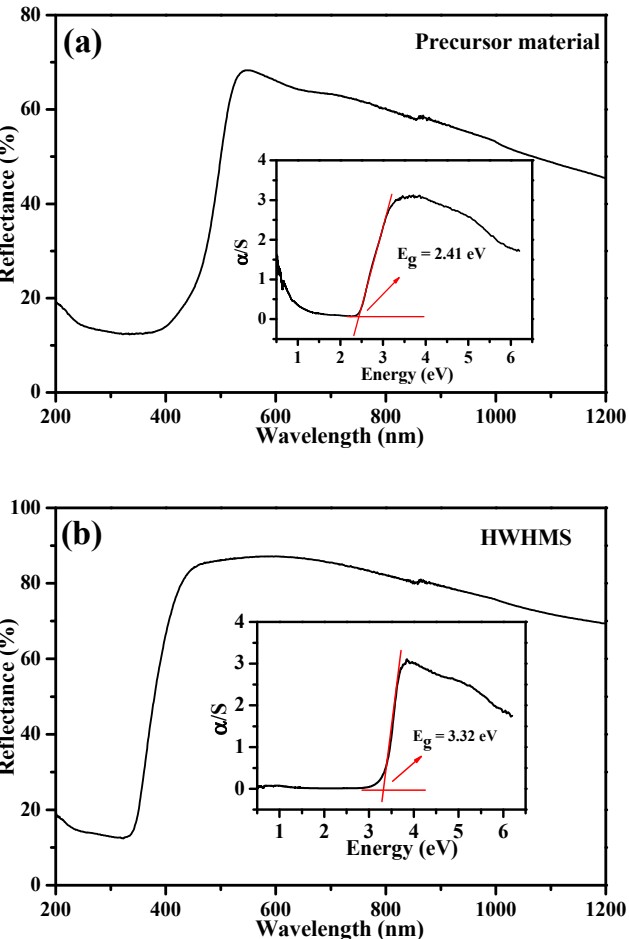

**Figure 5.** UV–Vis absorption spectrum and calculated band-gap diagram of (**a**) the precursor material and (**b**) the HWHMS.

### 2.3. Photocatalytic Performance of the HWHMS

It is well accepted that the size and morphology of photocatalysts have a great influence on their photocatalytic properties [30,31]. To investigate application of the HWHMS in the field of water pollution, photocatalytic degradation of MB by the HWHMS was performed under irradiation by a 300 W Xe lamp within the visible region (420–780 nm). The time-dependent absorption spectra of the MB solutions are shown in Figure 6a. Photocatalytic degradation of MB by isolated $WO_3$ nanorods was also performed for comparison (Figure 6b). The photocatalytic degradation performance was evaluated by comparing the intensity of MB at the absorption peak position (664 nm). The curve corresponding to 0 min is the absorption curve of MB, which had achieved dark adsorption equilibrium. The intensity of the characteristic adsorption peak decreased with increasing photocatalytic degradation time, indicating an effective degradation process in both cases. The plots of $C/C_0$ versus irradiation time for the HWHMS and $WO_3$ nanorods are shown in Figure 6c, where $C$ is the MB concentration at time t and $C_0$ is the initial MB concentration when adsorption–desorption equilibrium was achieved. After 70 min photodegradation, the adsorption ratios of the $WO_3$ nanorods and HWHMS were 58% and 80%, respectively, showing the higher photocatalytic performance of the as-synthesized HWHMS than the

isolated $WO_3 \cdot 0.33H_2O$ nanorods. Although HWHMS is not as good as other oxides in its photocatalytic effect, such as $ZnO/TiO_2$ nanosheets [32], compared with $WO_3 \cdot 0.33H_2O$ prepared by others in the literature, its photocatalytic effect is still relatively good, such as $WO_3 \cdot 0.33H_2O$ Nanonetworks [33], $Ag_2O/WO_3 \cdot 0.33H_2O$ heterostructure [34], $CeO_2$ Nanostructures [35], ZnO Nanowires [36] and so on.

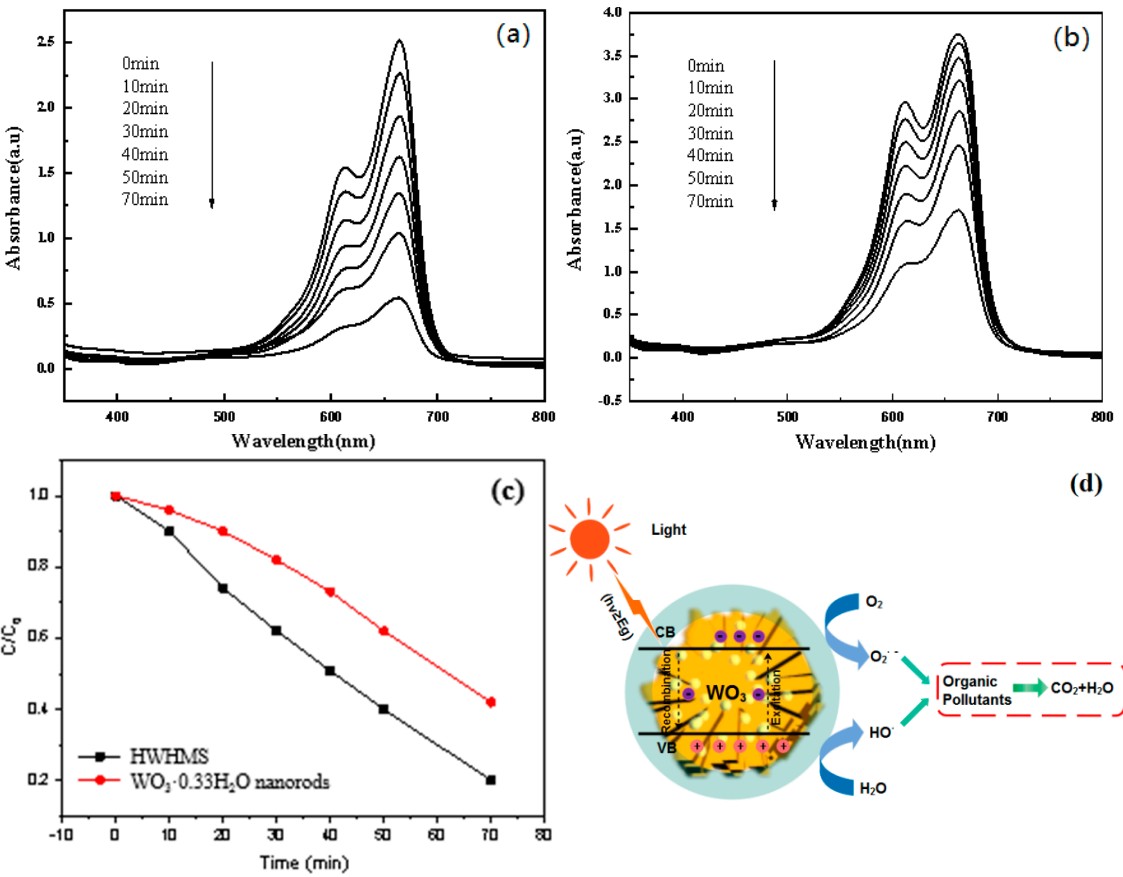

**Figure 6.** UV–Vis spectra of MB with different illumination times under visible light in the (**a**) HWHMS and (**b**) $WO_3 \cdot 0.33H_2O$ nanorods. (**c**) Photocatalytic degradation rate of MB (30 mg/L) by the HWHMS and $WO_3$ nanorods. (**d**) Schematic diagram MB degradation by the HWHMS.

The difference in the activities of the $WO_3$ nanorods and HWHMS can be ascribed to the following factors. First, the narrower bandgap of the HWHMS would contribute to higher degradation efficiency. Generally, the energy band structure of nanostructured $WO_3$ consists of a discontinuous low-energy valence band (VB) full of electrons and an empty high-energy conduction band (CB), which are separated by the band gap. When light with energy equal to or greater than the band gap of $WO_3$ hits the surface, a VB electron will absorb the light energy to excite transition to the CB, generating an electron–hole pair. The electron reacts with a neighboring $O_2$ molecules to produce a superoxide radical ($\cdot O_2^-$), and the hole simultaneously reacts with a water molecule on the photocatalyst surface to produce a highly reactive hydroxyl radical ($\cdot OH$). The $\cdot O_2^-$ and OH radicals can oxidize the MB molecules adsorbed on the surface of the $WO_3$ catalyst particles to harmless substances. A schematic of the mechanism of electron–hole generation, radical formation, and reaction with organic dyes in water for degradation is shown in Figure 6d. The narrower band gap of the HWHMS means that lower irradiated energy is required to generate an electron–hole pair and more light can be absorbed. Furthermore, the narrow band gap can also prevent recombination of photogenerated electron–hole pairs. Hence, catalytic degradation of organic pollutants by the HWHMS is enhanced compared with the isolated $WO_3 \cdot 0.33H_2O$ nanorods.

Second, it is generally accepted that the catalytic process is mainly related to adsorption and desorption of molecules on the surface of the catalyst. The calculated BET surface area of the $WO_3 \cdot 0.33H_2O$ nanorods is 9.7993 $m^2/g$. The high specific surface area of the HWHMS composed of stacking nanorods results in more unsaturated surface coordination sites exposed to the solution. The hierarchical structures in the HWHMS catalyst enable storage of more molecules. Therefore, the nature of the HWHMS structure provides more active reaction sites for the MB degradation reaction, leading to improved photoactivity.

Finally, different crystal phases also influence the photocatalytic activity of $WO_3$. The HWHMS is the orthorhombic $WO_3$ phase, while our previous work indicates that $WO_3 \cdot 0.33H_2O$ nanorods are the hexagonal $WO_3$ phase [37]. Specifically, the $WO_3 \cdot 0.33H_2O$ crystal structure contains two types of $WO_3$ octahedrons. One contains six W–O bonds and the other contains four W–O bonds, a shorter W=O bond, and a longer $W–OH_2$ bond [38,39]. In the orthorhombic phase structure, W=O in the upper layer and $W–OH_2$ in the lower layer cannot form hydrogen bonds. Conversely in the hexagonal phase structure, hydrogen bonds can easily form between W=O and $W–OH_2$ in the adjacent lower layer. This structural difference enables the orthorhombic $WO_3$ photocatalyst to maintain a shorter W=O bond, which makes the Fermi level and CB position of the orthorhombic $WO_3$ phase higher than those of the hexagonal $WO_3$ phase. The transition to the CB is thus easier, inhibiting recombination of photogenerated electron–hole pairs, so the photocatalytic degradation performance of the orthorhombic $WO_3$ phase is higher than that of the hexagonal $WO_3$ phase.

Generally, two steps are involved in photodegradation of organic dyes: adsorption of the dye molecules and their subsequent degradation. The pseudo-first-order and pseudo-second-order kinetics models were used to further investigate the adsorption mechanism associated with the HWHMS. The pseudo-first-order kinetics model can be expressed as [40]

$$\frac{1}{q_t} = \frac{K_1}{tq_e} + \frac{1}{q_e}, \ldots \tag{2}$$

while the pseudo-second-order kinetics model can be expressed as [35]

$$\frac{t}{q_t} = \frac{1}{k_2 q_e^2} + \left(\frac{1}{q_e}\right)t, \tag{3}$$

where $q_e$ is the absorption capacity at equilibrium and $q_t$ is the loading of MB at time $t$. The parameters $k_1$ and $k_2$ represent the pseudo-first-order and pseudo-second-order rate constants of the kinetics models, respectively. The linear fits of the data obtained using the HWHMS according to the above two models are shown in Figure 7a,b. The correlation coefficient ($R_2$) of the pseudo-second-order model (0.97095) was higher than that of the pseudo-first-order model (0.90711), suggesting that the HWHMS tended to adsorb MB by chemical processes, and that adsorption was dependent on both the surface properties of the adsorbent and the solute concentration.

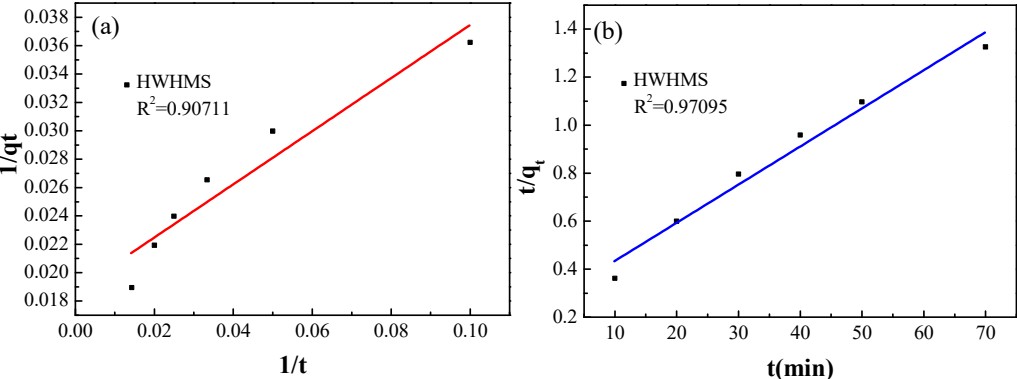

**Figure 7.** (**a**) Pseudo-first-order and (**b**) pseudo-second-order kinetic models for MB degradation by the HWHMS.

## 3. Materials and Methods

### 3.1. Preparation of the HWHMS

All of the reagents used in the experiments were obtained from commercial sources, and they were analytical reagent grade and used without further purification. First, $WO_3$ seeds were prepared. Specifically, 3 mM $Na_2WO_4 \cdot 2H_2O$ was dissolved in 20 mL of deionized water with magnetic stirring to ensure that it completely dissolved, and then 3 M HCl was added to the solution dropwise until the solution began to produce a yellowish precipitate. After magnetic stirring for a few minutes, the precipitate was centrifuged at a speed of 4000 rpm for 5 min. Second, the HWHMS was synthesized by a hydrothermal process using water and hydrogen peroxide as the solvent. Typically, the resulting yellowish precipitate was dissolved in 15 mL of hydrogen peroxide aqueous solution (30 wt%). After stirring for 30 min, the resultant mixture was transferred into a Teflon-lined autoclave (50 mL) and reacted at 160 °C for 12 h. The mixture was then naturally cooled to room temperature. The resultant mixture was washed with deionized water and ethanol several times. Finally, the white precipitate was placed in a vacuum oven and dried at 60 °C for 1 day to acquire the final sample. For comparison, isolated $WO_3 \cdot 0.33H_2O$ nanorods were also prepared. The specific synthesis process is described in [37].

### 3.2. Characterization

The synthesized HWHMS was characterized by XRD (Bruker D8-Advance X-ray diffractometer, Bruker, Karlsruhe, Germany) from 10° to 70° at a rate of 4° (2θ) min$^{-1}$ at 40 kV and 30 mA using Cu Kα as the radiation source (λ = 1.5418 Å). FESEM (NOVA NANOSEM 450, FEI, Waltham, MA, USA) was performed to determine the morphology of the HWHMS. The composition and structures of the HWHMS were characterized by Raman spectroscopy (Renishaw Invia, Waltham, MA, USA) at a wavelength of 514.5 nm and TGA (NETZSCH STA 449 F3 thermal analyzer, NETZSCH-Gerätebau GmbH, Selbu, Germany), respectively. The BET specific surface area and pore structures of the HWHMS were investigated at 77 K on the basis of the nitrogen-adsorption isotherms obtained with a surface area and porosimetry analyzer (Micromeritics 3Flex surface area and porosimetry analyzer, Micromeritics, Norcross, GA, USA). The samples were degassed at 120 °C for 10 h before the measurements. The UV/Vis/NIR optical diffuse reflectance spectra were recorded at room temperature on a Perkin–Elmer Lambda 950 Spectrophotometer (Perkin–Elmer, Waltham, MA, USA) with a wavelength range of l = 190–2500 nm. A $BaSO_4$ plate was used as a standard (100% reflectance). The optical properties of the HWHMS were characterized by UV–Vis spectroscopy (UV-2600, Shimadzu, Kyoto, Japan). The sample was placed in a quartz cell with an optical path length of 1 cm.

### 3.3. Photocatalysis Experiments

The photocatalytic activity of the HWHMS was determined by degradation of MB with irradiation of visible light under ambient temperature. In the experiment, 0.075 g of HWHMS powder was dissolved in a beaker containing 50 mL (30 mg/L) MB solution. Before photocatalysis, the sample was placed in a completely dark environment for 30 min, which is referred to as "0". Subsequently, 5 mL of the suspension was extracted from the beaker and dropped into a centrifuge tube. The residual mixture was placed under a Xe lamp (CEL-HXF 300, 300 W, Zhongjiao Gold Source Co., Ltd., Beijing, China) for irradiation. The filter filtered the beam and guaranteed that the wavelength was in the visible band (λ = 420–780 nm). The illumination time was controlled to 210 min during the whole photocatalytic process, and 5 mL solution was added into a centrifuge tube every 30 min. The entire experiment was performed under magnetic stirring at a speed of 400 rpm and the distance between the light source and liquid level was maintained at 12 mm. The above solution taken at different times was centrifuged at 6000 rpm and the supernatant liquid was used to investigate the UV–Vis absorption properties.

## 4. Conclusions

The well-defined HWHMS was successfully prepared through self-assembly of $WO_3 \cdot 0.33H_2O$ nanorods by the hydrothermal method. The hexagonal morphology of the HWHMS was confirmed by FESEM, and XRD, Raman spectroscopy, and TGA showed that the synthesized product was orthorhombic $WO_3 \cdot 0.33H_2O$. Owing to the unique hierarchical microstructure, the HWHMS had larger BET surface and narrower bandgap (1.53 eV) than the isolated $WO_3 \cdot 0.33H_2O$ nanorods, which is beneficial for photocatalytic degradation of MB. The HWHMS showed enhanced photocatalytic activity for degradation of MB under visible-light irradiation compared with the isolated $WO_3 \cdot 0.33H_2O$ nanorods, because of its narrower bandgap, larger BET specific surface area, and orthorhombic phase structure. In addition, the HWHMS tended to adsorb MB through chemical processes. Experiments indicated that the surface area and crystal phase of the catalyst had a great influence on the photocatalytic activity. This type of novel superstructure is also expected to have potential applications in catalyst supports, solar cells, and gas sensors.

**Author Contributions:** C.Z. conceived the idea, initiated and supervised the project, discussed the results, and edited the manuscript. W.L. and T.W. designed and conducted the experiments, analyzed the results, co-wrote the manuscript, and contributed equally to the project. D.H. and X.X. guided the experiments of magnetic actuation and analyzed concerned results. S.C. and Y.L. conducted the calculation of bandgap. W.C. edited the manuscript. All authors have read and agreed to the published version of the manuscript.

**Funding:** This research was funded by the Natural Science Foundation of Fujian Province (grant number 2020J01896), the Central Guidance for Local Science and Technology Development Project (grant number 2018L3001), the Youth Natural Fund Key Project of Fujian Province (grant number JZ160462), the Major Project of the University of Fujian Province (grant number 2018H6001), and the Doctoral Fund of Fujian University of Technology (grant number GY-Z15002 and GY-Z15008).

**Data Availability Statement:** The data presented in this study are available on request from the corresponding author.

**Conflicts of Interest:** The authors declare no conflict of interest. The funders had no role in the design of the study; in the collection, analyses, or interpretation of data; in the writing of the manuscript, or in the decision to publish the results.

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
