# Peer review of "Hexagonal WO3·0.33H2O Hierarchical Microstructure with Efficient Photocatalytic Degradation Activity"

_catalysts, doi:10.3390/catal11040496_

Round 1

Reviewer 1 Report

The objective of the paper, which is to demonstrate the advantage of a hexagonal self-arranged nanorod structure of WO3 over photocatalytic properties is interesting.

There are two points which would require a little more precision.

  • On the optical properties of the materials, the authors offer an explanation for an indirect to direct gap change due to the arrangement and presence of chlorine ions.

This phenomenon has not been demonstrated, and no reference is offered to support this theory with respect to other systems. It might be necessary to detail a little more this part which is important for the properties of the material.

(2) In figure 7a, the proposed linear regression does not stick very well with the experimental points. It feels like the end point is a problem. By removing it, we would fall back on kinetics close to that of order 2. I have the feeling that the authors showed the results which suited them the most in relation to the conclusions they wanted to draw, namely adsorption by chemical process. It is not explained either how the study of the kinetic models of order 1 and of order 2 make it possible to determine the state of absorption of MB. A small paragraph to explain the purpose of this kinetic study would be interesting.

Author Response

Response to Reviewer 2 Comments

General comments: The objective of the paper, which is to demonstrate the advantage of a hexagonal self-arranged nanorod structure of WO3 over photocatalytic properties is interesting. There are two points which would require a little more precision.

Response: Thanks for your positive comments and constructive suggestion for our work.

Point 1: On the optical properties of the materials, the authors offer an explanation for an indirect to direct gap change due to the arrangement and presence of chlorine ions. This phenomenon has not been demonstrated, and no reference is offered to support this theory with respect to other systems. It might be necessary to detail a little more this part which is important for the properties of the material.

Response 1: Thanks for your constructive comments. The explanation of indirect or direct changes in the gap caused by the arrangement and existence of chloride ions in the paper is not consistent with the experimental results obtained from the test, so the previous explanation and conclusion is not reasonable to some extent. So we have changed the claim of our conclusion in the revised manuscript.

Point 2: In figure 7a, the proposed linear regression does not stick very well with the experimental points. It feels like the end point is a problem. By removing it, we would fall back on kinetics close to that of order 2. I have the feeling that the authors showed the results which suited them the most in relation to the conclusions they wanted to draw, namely adsorption by chemical process. It is not explained either how the study of the kinetic models of order 1 and of order 2 make it possible to determine the state of absorption of MB. A small paragraph to explain the purpose of this kinetic study would be interesting.

Response 2: Thanks for your kind comment and good suggestion. Pseudo-first-order kinetics and pseudo-second-order kinetics are classical models for studying adsorption kinetics, which are mainly used to determine the rate control steps of material transfer and physico-chemical reaction in the adsorption process. Among them, the pseudo-first-order kinetics is based on the assumed adsorption and diffusion step control, which belongs to the physical process. The pseudo-second-order kinetics assumes that the adsorption rate is determined by the square value of the number of unopended adsorption vacancies on the surface of the adsorbent, and that the adsorption process is controlled by the chemisorption mechanism, which involves electron sharing or electron transfer between the adsorbent and the adsorbent.

In the natural state, the kinetic removal mechanism of MB mainly exists in the form of physical adsorption and oxidative degradation. In order to explore the kinetic mechanism of WO3·0.33H2O in the removal process of MB, the pseudo-first-order kinetic equation and the pseudo-second-order kinetic equation were used to simulate the process. Through the correlation coefficient R2 obtained from the simulation of the two equations, it can be known that the correlation coefficient of the pseudo-second-order kinetic equation is greater than the pseudo-first-order kinetic equation, indicating that the removal process of MB follows the pseudo-second-order kinetic equation, and the removal of MB belongs to a chemical process, namely oxidative degradation.

Reviewer 2 Report

The authors describe in their manuscript entitled “Self-assembled hexagonal WO3·0.33H2O hierarchical microstructure with efficient photocatalytic degradation activity” the synthesis, characterization and testing of micrometer scaled WO3. Although the manuscript is basically written in a logical way, it needs a major revision prior being evaluated again for acceptance in this journal. The overall scientific content of the manuscript is rather low, but when thoroughly reworked, it could be accepted in the journal. The most important issues to be corrected are:

  1. The authors must provide and included XRD results of the precursor material as well as the literature comparison in Figure 1. Furthermore, the authors should investigate crystallite sizes from XRD data in comparison to SEM results. It seems, that the crystallite are far bigger than 100 nm, hence Scherrer equation should not be used for sizes determinations.
  2. The authors must provide SEM as well as TGA data of the precursor material, too. Furthermore, additional elemental analyses are needed to ensure assumptions made by the authors (e.g. W-, H-, Cl-contents).
  3. The optical investigations are highly doubtful, due to the fact that the authors claim the final product to be colorless (“white”), which is in complete conflict with the estimated band gap of 1.5 eV (ca. 820 nm). It seems obvious, that the author’s band gap calculation is erroneous, which would overcome the contradiction of literature results, too. Furthermore, the authors must precisely explain their assumption of the role of Cl ions in the change of gap transitions.
  4. The photocatalytic investigations are of poor quality, too. The authors must include comparative results of the same material from literature as well as results of other semiconductor materials which show high efficiency towards degradation of methylene blue.

Author Response

Response to Reviewer 1 Comments

General comments:

The authors describe in their manuscript entitled “Hexagonal WO3·0.33H2O hierarchical microstructure with efficient photocatalytic degradation activity” the synthesis, characterization and testing of micrometer scaled WO3. Although the manuscript is basically written in a logical way, it needs a major revision prior being evaluated again for acceptance in this journal. The overall scientific content of the manuscript is rather low, but when thoroughly reworked, it could be accepted in the journal. The most important issues to be corrected are:

Point 1: The authors must provide and included XRD results of the precursor material as well as the literature comparison in Figure 1. Furthermore, the authors should investigate crystallite sizes from XRD data in comparison to SEM results. It seems, that the crystallite are far bigger than 100 nm, hence Scherrer equation should not be used for sizes determinations.

Response 1: We thank the referee for the suggestions. In the revised paper, we have added the XRD results of the precursor material, by comparing PDF standard card, the main substance of precursor was determined to be WO3·H2O(PDF# 43-0679), showed in Fig. 1. The diffraction peaks of the as-synthesized HWHMS can be exclusively indexed to orthorhombic WO3·0.33H2O (JCPDS Card No. 54-1012) with lattice constants of a = 12.5271 Å, b = 7.7367 Å, and c = 7.3447 Å.

Supplementary Fig. 1. XRD pattern of as-synthesized HWHMS and the precursor material.

Point 2: The authors must provide SEM as well as TGA data of the precursor material, too. Furthermore, additional elemental analyses are needed to ensure assumptions made by the authors (e.g. W-, H-, Cl-contents).

Response 2: We added the SEM and TGA data of the precursor material showed in Fig. 2 and Fig. 3 respectively. The average size of a single precursor material was approximately 1 μm. The weight loss process of the precursor material can be divided into three stages, occurred from room temperature to 800 °C with the weight loss of 12.39 %. The first slight weight loss stage (about 7.38 %) of the sample occurred between 0 to 300 °C, which is attributed to evaporation of free water from precursor material. The second slight weight loss stage (about 1.29 %) of the sample occurred between 300 to 600 °C, which is attributed to thermal decomposition of solvents from the precursor material. The third weight loss stage occurred from 600 to 800 °C the weightlessness of the third stage is 3.72 %, and it was probably associated with the loss of combined water. The weight loss rate of the precursors was more pronounced than that of HWHMS.

Supplementary Fig. 2. Typical SEM images of the precursor material

Supplementary Fig. 3. TGA curve of HWHMS and the precursor material

Point 3: The optical investigations are highly doubtful, due to the fact that the authors claim the final product to be colorless (“white”), which is in complete conflict with the estimated band gap of 1.5 eV (ca. 820 nm). It seems obvious, that the author’s band gap calculation is erroneous, which would overcome the contradiction of literature results, too. Furthermore, the authors must precisely explain their assumption of the role of Cl ions in the change of gap transitions.

Response 3: Thanks for your constructive comments. The explanation of indirect or direct changes in the gap caused by the arrangement and existence of chloride ions in the paper is not consistent with the experimental results obtained from the test, so the previous explanation and conclusion is not reasonable to some extent. So we have changed the claim of our conclusion in the revised manuscript.

For WO3·0.33H2O the transition is indirect. To further explore its band structure, the band-gap energies can be estimated from a plot of (αhν)1/2 versus photo-energy( hν) and the intercept of the tangent to the plot will give an approximation of the indirect band-gap energies of the samples. The plots of (αhν)1/2 versus hν are showed in Figure 4. The band gap energies of the precursor material and the HWHMS are calculated to be 2. 41 eV and 3.32 eV.

Supplementary Fig. 4. UV–Vis absorption spectrum and calculated band-gap diagram of (a) the precursor material; (b) the HWHMS.

Point 4: The photocatalytic investigations are of poor quality, too. The authors must include comparative results of the same material from literature as well as results of other semiconductor materials which show high efficiency towards degradation of methylene blue.

Response 4: Although HWHMS is not as good as other oxides in its photocatalytic effect, such as ZnO/TiO2 nanosheets[35], its photocatalytic effect is still relatively good compared with WO3·0.33H2O sample reported by others, such as WO3·0.33H2O Nanonetworks[32], Ag2O/WO3·0.33H2O heterostructure[33], CeO2 Nanostructures[34], ZnO Nanowires[36]. The following Table 1 summarizes the photocatalytic ability comparison with other counterparts.

Sample

Degradation efficiency

Reference

WO3·0.33H2O Nanorods

58% in 70 min

Our work

HWHMS

80% in 70 min

WO3·0.33H2O Nanonetworks

80% in 5 h

[32]He, X.; Hu, C.; Yi, Q.; Xue, W.; Hao, H.; Li, X. Preparation and Improved Photocatalytic Activity of WO3·0.33H2O Nanonetworks. Catal. Lett. 2012, 142, 637-645.

Ag2O/WO3·0.33H2O heterostructure

80% in 100 min

[33]He, X.Y.; Hu, C.; Xi, Y.; Zhang, K.Y.; Hua, H. Three-dimensional Ag2O/WO3·0.33H2O heterostructures for improving photocatalytic activity. Mater. Res. Bull. 2014, 50, 91-94.

CeO2 Nanostructures

70% in 150 min

[34]Deblina, M. I. C.; Kalyan, M.; Somenath, R., Facet-Dependent Photodegradation of Methylene Blue Using Pristine CeO2 Nanostructures. ACS Omega 2019, 4, 4243-4251.

ZnO/TiO2 nanosheets

80% in 40 min

[35]Wang, L.; Liu, S.; Wang, Z.; Zhou, Y.; Qin, Y.; Wang, Z.L. Piezotronic Effect Enhanced Photocatalysis in Strained Anisotropic ZnO/TiO2 Nanoplatelets via Thermal Stress. ACS Nano 2016, 10, 2636-2643.

ZnO Nanowires

80% in 75 min

[36]Smazna, D.; Shree, S.; Polonskyi, O.; Mutual Interplay of ZnO Micro- and Nanowires and Methylene Blue during Cyclic Photocatalysis Process. J. Environ. Chem. Eng. 2019, 7, 103016.

Round 2

Reviewer 2 Report

The authors answered the issues to my satisfaction. Hence, I can support acceptance of the manuscript.